# Hermite–Hadamard's Integral Inequalities of $(\alpha, s)$-GA- and $(\alpha, s, m)$-GA-Convex Functions

**Jing-Yu Wang** [1], **Hong-Ping Yin** [1], **Wen-Long Sun** [2] **and Bai-Ni Guo** [3,4,*]

1   College of Mathematics and Physics, Inner Mongolia Minzu University, Tongliao 028043, China
2   Department of Mathematics, School of Science, Shenyang University of Technology, Shenyang 110870, China
3   School of Mathematics and Informatics, Henan Polytechnic University, Jiaozuo 454003, China
4   Independent Researcher, Dallas, TX 75252-8024, USA
*   Correspondence: bai.ni.guo@gmail.com

**Abstract:** In this paper, the authors propose the notions of $(\alpha, s)$-geometric-arithmetically convex functions and $(\alpha, s, m)$-geometric-arithmetically convex functions, while they establish some new integral inequalities of the Hermite–Hadamard type for $(\alpha, s)$-geometric-arithmetically convex functions and for $(\alpha, s, m)$-geometric-arithmetically convex functions.

**Keywords:** Hermite–Hadamard type integral inequality; $(\alpha, s)$-geometric-arithmetically convex function; $(\alpha, s, m)$-geometric-arithmetically convex function

**MSC:** Primary 26A51; Secondary 26D15; 41A55

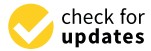



## 1. Introduction

In this paper, we denote a nonempty and open interval with $I \subseteq \mathbb{R}$.

We first review some definitions of various convex functions and list some Hermite–Hadamard-type integral inequalities.

It is general knowledge that a function $f : I \subseteq \mathbb{R} \to \mathbb{R}$ is said to be convex if

$$f(tx + (1-t)y) \leq tf(x) + (1-t)f(y)$$

for all $x, y \in I$ and $t \in [0, 1]$. One can find a lot of classical conclusions for convex functions in monographs [1,2].

In [3], Xi and his co-authors defined $(\alpha, s)$-convex functions and $(\alpha, s, m)$-convex functions and established some Hermite–Hadamard-type integral inequalities.

**Definition 1** ([3]). *For some $s \in [-1, 1]$ and $\alpha \in (0, 1]$, a function $f : I \subseteq \mathbb{R} \to \mathbb{R}$ is said to be $(\alpha, s)$-convex if*

$$f(tx + (1-t)y) \leq t^{\alpha s} f(x) + (1 - t^{\alpha})^s f(y)$$

*holds for all $x, y \in I$ and $t \in (0, 1)$.*

**Definition 2** ([3]). *For some $s \in [-1, 1]$ and $(\alpha, m) \in (0, 1] \times (0, 1]$, a function $f : [0, b] \to \mathbb{R}$ is said to be $(\alpha, s, m)$-convex if*

$$f(tx + m(1-t)y) \leq t^{\alpha s} f(x) + m(1 - t^{\alpha})^s f(y)$$

*holds for all $x, y \in [0, b]$ and $t \in (0, 1)$.*

**Definition 3** ([4,5]). *The function $f : I \subseteq \mathbb{R}_+ = (0, \infty) \to \mathbb{R}$ is said to be geometric-arithmetically convex, that is, GA-convex, on $I$ if*

$$f(x^t y^{1-t}) \leq tf(x) + (1-t)f(y)$$

*holds for all $x, y \in I$ and $t \in [0, 1]$.*

In [6], Shuang and her co-authors, including the second author of this paper, introduced the notion of the geometric-arithmetically $s$-convex function and established some inequalities of the Hermite–Hadamard type for geometric-arithmetically $s$-convex functions.

**Definition 4** ([6]). *Let $f : I \subseteq \mathbb{R}_+ \to \mathbb{R}_0 = [0, \infty)$ and $s \in (0, 1]$. A function $f(x)$ is said to be geometric-arithmetically $s$-convex on $I$ if*

$$f\left(x^t y^{1-t}\right) \leq t^s f(x) + (1 - t)^s f(y)$$

*holds for all $x, y \in I$ and $t \in (0, 1]$.*

**Remark 1.** *When $s = 1$, a geometric-arithmetically $s$-convex function becomes the GA-convex function defined in [4,5].*

**Remark 2.** *The integral estimates and applications of geometric-arithmetically convex functions have received renewed attention in recent years. A remarkable variety of refinements and generalizations have been found in, for example, [3–6]. In this paper, we will generalize the results of the above-mentioned literature and study the application problems.*

Let $f : I \subseteq \mathbb{R} \to \mathbb{R}$ be a convex function on $I$. Then, the Hermite–Hadamard integral inequality reads that

$$f\left(\frac{x + y}{2}\right) \leq \frac{1}{y - x} \int_x^y f(x) \, \mathrm{d}x \leq \frac{f(x) + f(y)}{2}, \quad x, y \in I.$$

One can find a lot of classical conclusions for the Hermite–Hadamard integral inequality in the monograph [7].

Hermite–Hadamard-type integral inequalities are a very active research topic [8]. We now recall some known results below.

**Theorem 1** ([9], Theorem 2.2). *Let $f : I^\circ \subseteq \mathbb{R} \to \mathbb{R}$ be a differentiable mapping on $I^\circ$, and let the points $a, b \in I^\circ$ with $a < b$. If $|f'|$ is convex on $[a, b]$, then*

$$\left| \frac{f(a) + f(b)}{2} - \frac{1}{b - a} \int_a^b f(x) \, \mathrm{d}x \right| \leq \frac{(b - a)(|f'(a)| + |f'(b)|)}{8}.$$

**Theorem 2** ([10], Theorems 1 and 2). *Let $f : I \subseteq \mathbb{R} \to \mathbb{R}$ be differentiable on $I^\circ$, and let $a, b \in I$ with $a < b$. If $|f'|^q$ is convex on $[a, b]$ for $q \geq 1$, then*

$$\left| \frac{f(a) + f(b)}{2} - \frac{1}{b - a} \int_a^b f(x) \, \mathrm{d}x \right| \leq \frac{b - a}{4} \left( \frac{|f'(a)|^q + |f'(b)|^q}{2} \right)^{1/q}$$

*and*

$$\left| f\left(\frac{a + b}{2}\right) - \frac{1}{b - a} \int_a^b f(x) \, \mathrm{d}x \right| \leq \frac{b - a}{4} \left( \frac{|f'(a)|^q + |f'(b)|^q}{2} \right)^{1/q}.$$

**Theorem 3** ([11]). *Let $f : \mathbb{R}_0 \to \mathbb{R}$ be $m$-convex and $m \in (0, 1]$. If $f \in L_1([a, b])$ for $0 \leq a < b < \infty$, then*

$$\frac{1}{b - a} \int_a^b f(x) \, \mathrm{d}x \leq \min\left\{ \frac{f(a) + mf(b/m)}{2}, \frac{mf(a/m) + f(b)}{2} \right\}.$$

**Theorem 4** ([12]). *Let $f : I \subseteq \mathbb{R}_0 \to \mathbb{R}$ be differentiable on $I^\circ$, the numbers $a, b \in I$ with $a < b$, and $f' \in L_1([a,b])$. If $|f'|^q$ is s-convex on $[a,b]$ for some fixed $s \in (0,1]$ and $q \geq 1$, then*

$$
\left| \frac{f(a) + f(b)}{2} - \frac{1}{b-a} \int_a^b f(x) \, \mathrm{d}x \right|
$$
$$
\leq \frac{b-a}{2} \left( \frac{1}{2} \right)^{1-1/q} \left[ \frac{2 + 1/2^s}{(s+1)(s+2)} \right]^{1/q} \left[ |f'(a)|^q + |f'(b)|^q \right]^{1/q}.
$$

**Theorem 5** ([13]). *Let $f : I \subseteq \mathbb{R}_0 \to \mathbb{R}$ be differentiable on $I^\circ$, let $a, b \in I$ with $a < b$, and let $f' \in L_1([a,b])$. If $|f'|^q$ is s-convex on $[a,b]$ for some fixed $s \in (0,1]$ and $q > 1$, then*

$$
\left| f\left( \frac{a+b}{2} \right) - \frac{1}{b-a} \int_a^b f(x) \, \mathrm{d}x \right| \leq \frac{b-a}{4} \left[ \frac{1}{(s+1)(s+2)} \right]^{1/q} \left( \frac{1}{2} \right)^{1/p} \left\{ \left[ |f'(a)|^q \right. \right.
$$
$$
\left. + (s+1) \left| f'\left( \frac{a+b}{2} \right) \right|^q \right]^{1/q} + \left[ |f'(b)|^q + (s+1) \left| f'\left( \frac{a+b}{2} \right) \right|^q \right]^{1/q} \right\},
$$

*where $\frac{1}{p} + \frac{1}{q} = 1$.*

**Theorem 6** ([14]). *Let $f : I \subseteq \mathbb{R}_0 \to \mathbb{R}$ be differentiable on $I^\circ$, let $a, b \in I$ with $a < b$, and let $f' \in L_1([a,b])$. If $|f'|$ is s-convex on $[a,b]$ for some $s \in (0,1]$, then*

$$
\left| \frac{1}{6} \left[ f(a) + 4f\left( \frac{a+b}{2} \right) + f(b) \right] - \frac{1}{b-a} \int_a^b f(x) \, \mathrm{d}x \right|
$$
$$
\leq \frac{(s-4)6^{s+1} + 2 \times 5^{s+2} - 2 \times 3^{s+2} + 2}{6^{s+2}(s+1)(s+2)} (b-a) \left( |f'(a)| + |f'(b)| \right).
$$

Motivated by the studies above, we will introduce the notions of "$(\alpha, s)$-geometric-arithmetically convex functions" and "$(\alpha, s, m)$-geometric-arithmetically convex functions", and we will establish some new inequalities of the Hermite–Hadamard type for $(\alpha, s)$-geometric-arithmetically convex functions and for $(\alpha, s, m)$-geometric-arithmetically convex functions.

## 2. Definitions

We now introduce the notions of "$(\alpha, s)$-geometric-arithmetically convex functions" and "$(\alpha, s, m)$-geometric-arithmetically convex functions".

**Definition 5.** *For some $s \in [-1, 1]$ and $\alpha \in (0, 1]$, a function $f : I \subseteq \mathbb{R}_+ \to \mathbb{R}$ is said to be $(\alpha, s)$-geometric-arithmetically convex, or simply speaking, $(\alpha, s)$-GA-convex if*

$$
f\left( x^t y^{1-t} \right) \leq t^{\alpha s} f(x) + (1 - t^\alpha)^s f(y)
$$

*holds for all $x, y \in I$ and $t \in (0, 1)$.*

**Remark 3.** *By Definition 5, we can see that,*
1. *If $\alpha = 1$, then $f(x)$ is an s-GA-convex function on $I$, see [6];*
2. *If $\alpha = s = 1$, then $f(x)$ is a GA-convex function on $I$, see [4,5].*

**Definition 6.** *For some $s \in [-1, 1]$ and $(\alpha, m) \in (0, 1] \times (0, 1]$, a function $f : (0, b] \subseteq \mathbb{R}_+ \to \mathbb{R}$ is said to be $(\alpha, s, m)$-geometric-arithmetically convex, or simply speaking, $(\alpha, s, m)$-GA-convex if*

$$
f\left( x^t y^{m(1-t)} \right) \leq t^{\alpha s} f(x) + m(1 - t^\alpha)^s f(y)
$$

*holds for all $x, y \in (0, b]$ and $t \in (0, 1)$.*

**Remark 4.** *By Definition 6, we can see that:*

1. *If $s = 1$, then $f(x)$ is an $(\alpha, m)$-GA-convex function on $(0, b]$;*
2. *If $\alpha = 1$, then $f(x)$ is an $(s, m)$-GA-convex function on $(0, b]$;*
3. *If $m = 1$, then $f(x)$ is an $(\alpha, s)$-GA-convex function on $(0, b]$.*

It is obvious that:

1. When $r \in (0, 1)$, the function $f(x) = x^r$ is strictly concave with respect to $x \in (0, 1]$;
2. When $r \in (-\infty, 0] \cup [1, \infty)$, the function $f(x) = x^r$ is convex with respect to $x \in (0, 1]$.

**Proposition 1.** *Let $\alpha \in (0, 1]$ and $s \in [-1, 0)$. Then, the function $f(x) = x^r$ for $r \in (0, 1)$ is $(\alpha, s)$-geometric-arithmetically convex with respect to $x \in \mathbb{R}_+$.*

**Proof.** We only need to verify the inequality

$$(x^r)^t (y^r)^{1-t} = f(x^t y^{1-t}) \leq t^{\alpha s} f(x) + (1 - t^\alpha)^s f(y) = t^{\alpha s} x^r + (1 - t^\alpha)^s y^r$$

for all $x, y \in \mathbb{R}$ and $t \in (0, 1)$.

For all $x, y \in \mathbb{R}$ and $t \in (0, 1)$:

1. When $x^r \leq y^r$, let $u = \frac{x^r}{y^r}$, then $0 \leq u \leq 1$ and $(1 - t^\alpha)^s > 1$; thus,

$$u^t \leq 1 < (1 - t^\alpha)^s < t^{\alpha s} u + (1 - t^\alpha)^s,$$

that is,

$$f(x^t y^{1-t}) \leq t^{\alpha s} f(x) + (1 - t^\alpha)^s f(y);$$

2. When $x^r \geq y^r$, we have

$$f(x^t y^{1-t}) = (x^r)^t (y^r)^{1-t} \leq x^r < t^{\alpha s} f(x) < t^{\alpha s} f(x) + (1 - t^\alpha)^s f(y).$$

The proof of Proposition 1 is complete. $\square$

## 3. Lemmas

The following lemmas are necessary for us.

**Lemma 1** ([15]). *Let $f : I \subseteq \mathbb{R} \to \mathbb{R}$ be differentiable on $I^\circ$ and let $a, b \in I$ with $a < b$. If $f' \in L_1([a, b])$, then for $x \in [a, b]$, we have*

$$\frac{(b - x) f(b) + (x - a) f(a)}{b - a} - \frac{1}{b - a} \int_a^b f(u) \, \mathrm{d} u$$
$$= \frac{(x - a)^2}{b - a} \int_0^1 (t - 1) f'(tx + (1 - t)a) \, \mathrm{d} t + \frac{(b - x)^2}{b - a} \int_0^1 (1 - t) f'(tx + (1 - t)b) \, \mathrm{d} t.$$

**Lemma 2.** *Let $\alpha \in (0, 1)$. Then,*

$$R_{-1}(\alpha) \triangleq \int_0^1 \frac{1 - t}{t^\alpha} \, \mathrm{d} t = \frac{1}{(1 - \alpha)(2 - \alpha)}$$

*and*

$$T_{-1}(\alpha) \triangleq \int_0^1 \frac{1 - t}{1 - t^\alpha} \, \mathrm{d} t = \frac{1}{\alpha} \left[ \psi\left(\frac{2}{\alpha}\right) - \psi\left(\frac{1}{\alpha}\right) \right],$$

*where $\psi(z) = \frac{\mathrm{d} \ln \Gamma(z)}{\mathrm{d} z}$, and*

$$\Gamma(z) = \int_0^1 t^{z-1} \mathrm{e}^{-t} \, \mathrm{d} t, \quad \Re(z) > 0$$

*denotes the classical Euler gamma function.*

**Proof.** By letting $u = t^\alpha$ for $t \in (0,1)$ and using the formulas

$$\psi(z) + \gamma = \int_0^1 \frac{1 - t^{z-1}}{1 - t} \, \mathrm{d}\, t$$

and

$$\gamma = \int_0^\infty \left( \frac{1}{1 + t} - \mathrm{e}^{-t} \right) \frac{\mathrm{d}\, t}{t}$$

in [16] (p. 259, 6.3.22), it is easy to show that

$$\int_0^1 \frac{1 - t}{1 - t^\alpha} \, \mathrm{d}\, t = \frac{1}{\alpha} \int_0^1 \frac{u^{1/\alpha - 1} - u^{2/\alpha - 1}}{1 - u} \, \mathrm{d}\, u = \frac{1}{\alpha} \left[ \psi\left(\frac{2}{\alpha}\right) - \psi\left(\frac{1}{\alpha}\right) \right].$$

The proof of Lemma 2 is complete. $\square$

### 4. Hermite–Hadamard-Type Integral Inequalities

In this section, we turn our attention to the establishment of integral inequalities of the Hermite–Hadamard type for $(\alpha, s)$-GA-convex and $(\alpha, s, m)$-GA-convex functions.

**Theorem 7.** *For some $s \in [-1, 1]$ and $\alpha \in (0, 1]$, let $f : I \subseteq \mathbb{R}_+ \to \mathbb{R}$ be a differentiable function on $I^\circ$, let $a, b \in I^\circ$ with $a < b$ and $x \in [a, b]$, and let $f' \in L_1([a, b])$ and $|f'|$ be decreasing on $[a, b]$. If $|f'|^q$ is $(\alpha, s)$-GA-convex on $[a, b]$ for $q \geq 1$, then the following conclusions are valid:*

1. *When $s \in (-1, 1]$ and $\alpha \in (0, 1]$, we have*

$$\left| \frac{(b - x)f(b) + (x - a)f(a)}{b - a} - \frac{1}{b - a} \int_a^b f(u) \, \mathrm{d}\, u \right|$$

$$\leq \left(\frac{1}{2}\right)^{1 - 1/q} \left\{ \frac{(x - a)^2}{b - a} \left[ R(\alpha, s)|f'(x)|^q + T(\alpha, s)|f'(a)|^q \right]^{1/q} \right.$$

$$\left. + \frac{(b - x)^2}{b - a} \left[ R(\alpha, s)|f'(x)|^q + T(\alpha, s)|f'(b)|^q \right]^{1/q} \right\}, \quad (1)$$

*where $R(\alpha, s)$ and $T(\alpha, s)$ are defined by*

$$R(\alpha, s) \triangleq \frac{1}{(\alpha s + 1)(\alpha s + 2)}$$

*and*

$$T(\alpha, s) \triangleq \frac{1}{\alpha} \left[ B\left(s + 1, \frac{1}{\alpha}\right) - B\left(s + 1, \frac{2}{\alpha}\right) \right]$$

*for $s \in (-1, 1]$;*

2. *When $s = -1$ and $\alpha \in (0, 1)$, we have*

$$\left| \frac{(b - x)f(b) + (x - a)f(a)}{b - a} - \frac{1}{b - a} \int_a^b f(u) \, \mathrm{d}\, u \right|$$

$$\leq \left(\frac{1}{2}\right)^{1 - 1/q} \left\{ \frac{(x - a)^2}{b - a} \left[ R_{-1}(\alpha)|f'(x)|^q + T_{-1}(\alpha)|f'(a)|^q \right]^{1/q} \right.$$

$$\left. + \frac{(b - x)^2}{b - a} \left[ R_{-1}(\alpha)|f'(x)|^q + T_{-1}(\alpha)|f'(b)|^q \right]^{1/q} \right\},$$

*where $R_{-1}(\alpha)$, $T_{-1}(\alpha)$ are defined in Lemma 2 and*

$$B(x,y) = \int_0^1 t^{x-1}(1-t)^{y-1}\, d\,t, \quad \Re(x), \Re(y) > 0 \tag{2}$$

*denotes the classical beta function.*

**Proof.** For $s \in (-1,1]$ and $\alpha \in (0,1]$, since $|f'|$ is decreasing on $[a,b]$, by Lemma 1 and the Hölder integral inequality, we have

$$
\begin{aligned}
&\left| \frac{(b-x)f(b) + (x-a)f(a)}{b-a} - \frac{1}{b-a}\int_a^b f(u)\, d\,u \right| \\
&\leq \frac{(x-a)^2}{b-a}\int_0^1 (1-t)|f'(tx+(1-t)a)|\, d\,t \\
&\quad + \frac{(b-x)^2}{b-a}\int_0^1 (1-t)|f'(tx+(1-t)b)|\, d\,t \\
&\leq \frac{(x-a)^2}{b-a}\int_0^1 (1-t)|f'(x^t a^{1-t})|\, d\,t + \frac{(b-x)^2}{b-a}\int_0^1 (1-t)|f'(x^t b^{1-t})|\, d\,t \\
&\leq \frac{(x-a)^2}{b-a}\left[\int_0^1 (1-t)\, d\,t\right]^{1-1/q}\left[\int_0^1 (1-t)|f'(x^t a^{1-t})|^q\, d\,t\right]^{1/q} \\
&\quad + \frac{(b-x)^2}{b-a}\left[\int_0^1 (1-t)\, d\,t\right]^{1-1/q}\left[\int_0^1 (1-t)|f'(x^t b^{1-t})|^q\, d\,t\right]^{1/q}.
\end{aligned}
\tag{3}
$$

Making use of the $(\alpha, s)$-GA-convexity of $|f'|^q$, we have

$$
\begin{aligned}
\int_0^1 (1-t)|f'(x^t a^{1-t})|^q\, d\,t &\leq \int_0^1 (1-t)\left[t^{\alpha s}|f'(x)|^q + (1-t^\alpha)^s|f'(a)|^q\right] d\,t \\
&= R(\alpha,s)|f'(x)|^q + T(\alpha,s)|f'(a)|^q
\end{aligned}
$$

and

$$
\begin{aligned}
\int_0^1 (1-t)|f'(x^t b^{1-t})|^q\, d\,t &\leq \int_0^1 (1-t)\left[t^{\alpha s}|f'(x)|^q + (1-t^\alpha)^s|f'(b)|^q\right] d\,t \\
&= R(\alpha,s)|f'(x)|^q + T(\alpha,s)|f'(b)|^q.
\end{aligned}
\tag{4}
$$

By using the above inequalities between (3) and (4) and then simplifying them, we obtain the required inequality (1).

When $s = -1$ and $\alpha \in (0,1)$, by the inequalities between (3) and (4) and by Lemma 2, we have

$$
\left| \frac{(b-x)f(b)+(x-a)f(a)}{b-a} - \frac{1}{b-a}\int_a^b f(u)\,\mathrm{d}u \right|
$$

$$
\leq \frac{(x-a)^2}{b-a}\left(\frac{1}{2}\right)^{1-1/q}\left[\int_0^1 (1-t)|f'(x^t a^{1-t})|^q\,\mathrm{d}t\right]^{1/q}
$$

$$
+ \frac{(b-x)^2}{b-a}\left(\frac{1}{2}\right)^{1-1/q}\left[\int_0^1 (1-t)|f'(x^t b^{1-t})|^q\,\mathrm{d}t\right]^{1/q}
$$

$$
\leq \frac{(x-a)^2}{b-a}\left(\frac{1}{2}\right)^{1-1/q}\left[\int_0^1 (1-t)\left[t^{-\alpha}|f'(x)|^q + (1-t^\alpha)^{-1}|f'(a)|^q\right]\mathrm{d}t\right]^{1/q}
$$

$$
+ \frac{(b-x)^2}{b-a}\left(\frac{1}{2}\right)^{1-1/q}\left[\int_0^1 (1-t)\left[t^{-\alpha}|f'(x)|^q + (1-t^\alpha)^{-1}|f'(b)|^q\right]\mathrm{d}t\right]^{1/q}
$$

$$
= \left(\frac{1}{2}\right)^{1-1/q}\left\{ \frac{(x-a)^2}{b-a}\left[R_{-1}(\alpha)|f'(x)|^q + T_{-1}(\alpha)|f'(a)|^q\right]^{1/q}\right.
$$

$$
\left. + \frac{(b-x)^2}{b-a}\left[R_{-1}(\alpha)|f'(x)|^q + T_{-1}(\alpha)|f'(b)|^q\right]^{1/q}\right\}.
$$

The proof of Theorem 7 is complete. □

In Theorem 7, when taking $\alpha = 1$ and $s \in (0,1]$, we derive the same result as in [6].

**Corollary 1.** *Under the conditions of Theorem 7, with $\alpha = 1$ and $s \in (-1,1]$, we have*

$$
\left| \frac{(b-x)f(b)+(x-a)f(a)}{b-a} - \frac{1}{b-a}\int_a^b f(u)\,\mathrm{d}u \right|
$$

$$
\leq \frac{(x-a)^2}{b-a}\left(\frac{1}{2}\right)^{1-1/q}\left[\frac{|f'(x)|^q + (s+1)|f'(a)|^q}{(s+1)(s+2)}\right]^{1/q}
$$

$$
+ \frac{(b-x)^2}{b-a}\left(\frac{1}{2}\right)^{1-1/q}\left[\frac{|f'(x)|^q + (s+1)|f'(b)|^q}{(s+1)(s+2)}\right]^{1/q}.
$$

In Theorem 7, when setting $\alpha = s = 1$, we deduce the following integral inequalities of the Hermite–Hadamard type for the GA-convex function.

**Corollary 2.** *Under the conditions of Theorem 7, with $\alpha = s = 1$, we have*

$$
\left| \frac{(b-x)f(b)+(x-a)f(a)}{b-a} - \frac{1}{b-a}\int_a^b f(u)\,\mathrm{d}u \right|
$$

$$
\leq \left(\frac{1}{2}\right)^{1-1/q}\left\{ \frac{(x-a)^2}{b-a}\left[\frac{|f'(x)|^q + 2|f'(a)|^q}{6}\right]^{1/q}\right.
$$

$$
\left. + \frac{(b-x)^2}{b-a}\left[\frac{|f'(x)|^q + 2|f'(b)|^q}{6}\right]^{1/q}\right\}.
$$

**Corollary 3.** *Under the conditions of Theorem 7, with $q = 1$ and $s \in (-1,1]$, we obtain*

$$
\left| \frac{(b-x)f(b)+(x-a)f(a)}{b-a} - \frac{1}{b-a}\int_a^b f(u)\,\mathrm{d}u \right|
$$

$$
\leq \frac{(x-a)^2}{b-a}\left[R(\alpha,s)|f'(x)| + T(\alpha,s)|f'(a)|\right]
$$

$$
+ \frac{(b-x)^2}{b-a}\left[R(\alpha,s)|f'(x)| + T(\alpha,s)|f'(b)|\right].
$$

By making use of the same method as that in the proof of Theorem 7, we obtain the following integral inequalities for $(\alpha, s, m)$-GA-convex functions.

**Theorem 8.** *For some fixed $(\alpha, m) \in (0, 1] \times (0, 1]$ and $s \in (-1, 1]$, let $a, b \in \mathbb{R}_+$ with $b > a$ and $x \in [a, b]$, let $f : (0, \max\{b, b^{1/m}\}] \to \mathbb{R}$ be a differentiable function, let $f' \in L_1([a, \max\{b, b^{1/m}\}])$, and let $|f'|$ be decreasing on $[a, b]$. If $|f'|^q$ is $(\alpha, s, m)$-GA-convex on $(0, \max\{b, b^{1/m}\}]$ for $q \geq 1$, then:*

1. *When $s \in (-1, 1]$ and $\alpha \in (0, 1]$, we have*

$$
\left| \frac{(b - x)f(b) + (x - a)f(a)}{b - a} - \frac{1}{b - a} \int_a^b f(u)\, \mathrm{d}u \right|
$$
$$
\leq \left( \frac{1}{2} \right)^{1 - 1/q} \left\{ \frac{(x - a)^2}{b - a} \left[ R(\alpha, s)|f'(x)|^q + mT(\alpha, s)|f'(a^{1/m})|^q \right]^{1/q} \right.
$$
$$
\left. + \frac{(b - x)^2}{b - a} \left[ R(\alpha, s)|f'(x)|^q + mT(\alpha, s)|f'(b^{1/m})|^q \right]^{1/q} \right\}; \quad (5)
$$

2. *When $s = -1$ and $\alpha \in (0, 1)$, we have*

$$
\left| \frac{(b - x)f(b) + (x - a)f(a)}{b - a} - \frac{1}{b - a} \int_a^b f(u)\, \mathrm{d}u \right|
$$
$$
\leq \left( \frac{1}{2} \right)^{1 - 1/q} \left\{ \frac{(x - a)^2}{b - a} \left[ R_{-1}(\alpha)|f'(x)|^q + mT_{-1}(\alpha)|f'(a^{1/m})|^q \right]^{1/q} \right.
$$
$$
\left. + \frac{(b - x)^2}{b - a} \left[ R_{-1}(\alpha)|f'(x)|^q + mT_{-1}(\alpha)|f'(b^{1/m})|^q \right]^{1/q} \right\}, \quad (6)
$$

*where $R(\alpha, s)$, $T(\alpha, s)$, $R_{-1}(\alpha)$, and $T_{-1}(\alpha)$ are defined respectively in Theorem 7 and Lemma 2.*

**Proof.** Using (3), we have

$$
\left| \frac{(b - x)f(b) + (x - a)f(a)}{b - a} - \frac{1}{b - a} \int_a^b f(u)\, \mathrm{d}u \right|
$$
$$
\leq \frac{(x - a)^2}{b - a} \left[ \int_0^1 (1 - t)\, \mathrm{d}t \right]^{1 - 1/q} \left[ \int_0^1 (1 - t)|f'(x^t a^{1-t})|^q\, \mathrm{d}t \right]^{1/q}
$$
$$
+ \frac{(b - x)^2}{b - a} \left[ \int_0^1 (1 - t)\, \mathrm{d}t \right]^{1 - 1/q} \left[ \int_0^1 (1 - t)|f'(x^t b^{1-t})|^q\, \mathrm{d}t \right]^{1/q}. \quad (7)
$$

Making use of the $(\alpha, s, m)$-GA-convexity of $|f'|^q$ on $(0, \max\{b, b^{1/m}\}]$ once again yields

$$
\int_0^1 (1 - t)|f'(x^t a^{1-t})|^q\, \mathrm{d}t = \int_0^1 (1 - t)|f'(x^t a^{m(1-t)/m})|^q\, \mathrm{d}t
$$
$$
\leq \int_0^1 \left[ (1 - t)t^{\alpha s}|f'(x)|^q + m(1 - t)(1 - t^\alpha)^s|f'(a^{1/m})|^q \right] \mathrm{d}t
$$
$$
= R(\alpha, s)|f'(x)|^q + mT(\alpha, s)|f'(a^{1/m})|^q
$$

and

$$
\int_0^1 (1 - t)|f'(x^t b^{1-t})|^q\, \mathrm{d}t \leq R(\alpha, s)|f'(x)|^q + mT(\alpha, s)|f'(b^{1/m})|^q.
$$

We then substitute the two inequalities above into (7) and simplify the result in the required inequality (5).

Similarly, we can prove inequality (6). The proof of Theorem 8 is complete. □

**Corollary 4.** *In Theorem 8, if $q = 1$ and $s \in (-1, 1]$, then*

$$
\left| \frac{(b-x)f(b) + (x-a)f(a)}{b-a} - \frac{1}{b-a} \int_a^b f(u)\,\mathrm{d}\,u \right|
$$

$$
\leq \frac{(x-a)^2}{b-a} \big[ R(\alpha, s) |f'(x)| + mT(\alpha, s) |f'(a^{1/m})| \big]
$$

$$
+ \frac{(b-x)^2}{b-a} \big[ R(\alpha, s) |f'(x)| + mT(\alpha, s) |f'(b^{1/m})| \big].
$$

**Theorem 9.** *For some fixed $(\alpha, m) \in (0, 1] \times (0, 1]$ and $s \in (-1, 1]$, let $a, b \in \mathbb{R}_+$ with $b > a$ and $x \in [a, b]$, let $f : (0, \max\{b, b^{1/m}\}] \to \mathbb{R}$ be a differentiable function, and let $f' \in L_1([a, \max\{b, b^{1/m}\}])$ and $|f'|$ be decreasing on $[a, b]$. If $|f'|^q$ is $(\alpha, s, m)$-GA-convex on $(0, \max\{b, b^{1/m}\}]$ for $q > 1$, then*

$$
\left| \frac{(b-x)f(b) + (x-a)f(a)}{b-a} - \frac{1}{b-a} \int_a^b f(u)\,\mathrm{d}\,u \right|
$$

$$
\leq \left( \frac{q-1}{2q-1} \right)^{1-1/q} \Bigg\{ \frac{(x-a)^2}{b-a} \left[ \frac{\alpha |f'(x)|^q + m(\alpha s + 1) B(\frac{1}{\alpha}, s+1) |f'(a^{1/m})|^q}{\alpha(\alpha s + 1)} \right]^{1/q}
$$

$$
+ \frac{(b-x)^2}{b-a} \left[ \frac{\alpha |f'(x)|^q + m(\alpha s + 1) B(\frac{1}{\alpha}, s+1) |f'(b^{1/m})|^q}{\alpha(\alpha s + 1)} \right]^{1/q} \Bigg\}, \quad (8)
$$

*where $B(x, y)$ is defined by (2) in Theorem 7.*

**Proof.** Since $|f'|$ is decreasing on $[a, b]$, by Lemma 1 and the Hölder integral inequality, we obtain

$$
\left| \frac{(b-x)f(b) + (x-a)f(a)}{b-a} - \frac{1}{b-a} \int_a^b f(u)\,\mathrm{d}\,u \right|
$$

$$
\leq \frac{(x-a)^2}{b-a} \left[ \int_0^1 (1-t)^{q/(q-1)}\,\mathrm{d}\,t \right]^{1-1/q} \left[ \int_0^1 |f'(x^t a^{1-t})|^q\,\mathrm{d}\,t \right]^{1/q}
$$

$$
+ \frac{(b-x)^2}{b-a} \left[ \int_0^1 (1-t)^{q/(q-1)}\,\mathrm{d}\,t \right]^{1-1/q} \left[ \int_0^1 |f'(x^t b^{1-t})|^q\,\mathrm{d}\,t \right]^{1/q}, \quad (9)
$$

where

$$
\int_0^1 (1-t)^{q/(q-1)}\,\mathrm{d}\,t = \frac{q-1}{2q-1},
$$

$$
\int_0^1 |f'(x^t a^{1-t})|^q\,\mathrm{d}\,t \leq \int_0^1 \left[ t^{\alpha s} |f'(x)|^q + m(1-t^\alpha)^s |f'(a^{1/m})|^q \right]\mathrm{d}\,t
$$

$$
= \frac{\alpha |f'(x)|^q + m(\alpha s + 1) B(\frac{1}{\alpha}, s+1) |f'(a^{1/m})|^q}{\alpha(\alpha s + 1)},
$$

and

$$
\int_0^1 |f'(x^t b^{1-t})|^q\,\mathrm{d}\,t \leq \frac{\alpha |f'(x)|^q + m(\alpha s + 1) B(\frac{1}{\alpha}, s+1) |f'(b^{1/m})|^q}{\alpha(\alpha s + 1)}.
$$

Note that in the above arguments, we used the fact that the function $|f'|^q$ is $(\alpha, s, m)$-GA-convex on $(0, \max\{b, b^{1/m}\}]$. Applying the above equality and inequalities into (9) and then simplifying them lead to the required inequality (8). The proof of Theorem 9 is complete. $\square$

Using the same method as that in the proof of Theorem 9, we obtain the following inequalities of $(\alpha, s)$-GA-convex functions.

**Theorem 10.** *For some $s \in (-1, 1]$ and $\alpha \in (0, 1]$, let $f : I \subseteq \mathbb{R}_+ \to \mathbb{R}$ be a differentiable function on $I^\circ$, let $a, b \in I^\circ$ with $a < b$ and $x \in [a, b]$, and let $f' \in L_1([a, b])$ and $|f'|$ be decreasing on $[a, b]$. If $|f'|^q$ is $(\alpha, s)$-GA-convex on $[a, b]$ for $q > 1$, then*

$$
\left| \frac{(b - x)f(b) + (x - a)f(a)}{b - a} - \frac{1}{b - a} \int_a^b f(u) \, du \right|
$$
$$
\leq \left( \frac{q - 1}{2q - 1} \right)^{1 - 1/q} \left\{ \frac{(x - a)^2}{b - a} \left[ \frac{\alpha |f'(x)|^q + (\alpha s + 1) B(\frac{1}{\alpha}, s + 1) |f'(a)|^q}{\alpha(\alpha s + 1)} \right]^{1/q} \right.
$$
$$
\left. + \frac{(b - x)^2}{b - a} \left[ \frac{\alpha |f'(x)|^q + (\alpha s + 1) B(\frac{1}{\alpha}, s + 1) |f'(b)|^q}{\alpha(\alpha s + 1)} \right]^{1/q} \right\},
$$

*where $B(x, y)$ is defined by (2) in Theorem 7.*

In Theorem 10, when $\alpha = 1$, the Hermite–Hadamard-type integral inequality is the same as the result in [6].

**Corollary 5** ([6]). *Under the conditions of Theorem 10, if we take $\alpha = 1$, then*

$$
\left| \frac{(b - x)f(b) + (x - a)f(a)}{b - a} - \frac{1}{b - a} \int_a^b f(u) \, du \right| \leq \left( \frac{q - 1}{2q - 1} \right)^{1 - 1/q}
$$
$$
\times \left[ \frac{(x - a)^2}{b - a} \left( \frac{|f'(x)|^q + |f'(a)|^q}{s + 1} \right)^{1/q} + \frac{(b - x)^2}{b - a} \left( \frac{|f'(x)|^q + |f'(b)|^q}{s + 1} \right)^{1/q} \right].
$$

## 5. Applications to Special Means

For two positive numbers $a, b \in \mathbb{R}_+$ with $b > a$, define

$$
A(a, b) = \frac{a + b}{2}, \quad H(a, b) = \frac{2ab}{a + b}, \quad L(a, b) = \frac{b - a}{\ln b - \ln a}
$$

and

$$
L_r(a, b) = \begin{cases} \left[ \dfrac{b^{r+1} - a^{r+1}}{(r + 1)(b - a)} \right]^{1/r}, & r \neq 0, -1; \\ L(a, b), & r = -1; \\ \dfrac{1}{e} \left( \dfrac{b^b}{a^a} \right)^{1/(b-a)}, & r = 0. \end{cases}
$$

These means are respectively called the arithmetic, harmonic, logarithmic, and generalized logarithmic means of $a, b \in \mathbb{R}_+$.

**Theorem 11.** *Let $a, b \in \mathbb{R}_+$ with $a < b$, let $0 \neq r \leq 1$, and let $q \geq 1$.*
1. *If $r \neq -1$, we have*

$$
|A(a^r, b^r) - L_r^r(a, b)|
$$
$$
\leq \frac{(b - a)|r|}{2} \left( \frac{1}{2} \right)^{2(1 - 1/q)} \left[ \left( \frac{2A^{(r-1)q}(a, b) + a^{(r-1)q}}{3} \right)^{1/q} \right.
$$
$$
\left. + \left( \frac{2A^{(r-1)q}(a, b) + b^{(r-1)q}}{3} \right)^{1/q} \right].
$$

2.　*If $r = -1$, we have*

$$\left| \frac{1}{H(a,b)} - \frac{1}{L(a,b)} \right|$$

$$\leq \frac{b-a}{2} \left( \frac{1}{2} \right)^{-2/q} \left[ \left( \frac{2A^{-2q}(a,b) + a^{-2q}}{3} \right)^{1/q} \right.$$

$$\left. + \left( \frac{2A^{-2q}(a,b) + b^{-2q}}{3} \right)^{1/q} \right].$$

**Proof.** In Corollary 1, let $x = \frac{a+b}{2}$ and $s = -\frac{1}{2}$. If $r \leq 1$ and $q \geq 1$, the $|f'(x)| = |r|x^{r-1}$ is decreasing on $[a,b]$. By Proposition 1, we can derive the inequalities in Theorem 11. □

**Corollary 6.** *Under the conditions of Theorem 11, with $q = 1$:*
1.　*If $r \neq -1$, we have*

$$|A(a^r, b^r) - L_r^r(a,b)| \leq (b-a)|r| \left[ \frac{a^{r-1} + 4[A(a,b)]^{r-1} + b^{r-1}}{6} \right];$$

2.　*If $r = -1$, we have*

$$\left| \frac{1}{H(a,b)} - \frac{1}{L(a,b)} \right| \leq (b-a)|r| \left[ \frac{a^{-2} + 4[A(a,b)]^{-2} + b^{-2}}{6} \right].$$

## 6. Conclusions

Integral inequalities are important for the prediction of upper and lower bounds in various aspects of applied sciences such as in Probability Theory, Functional Inequalities, and Information Theory.

In this paper, after recalling some convexities and the Hermite–Hadamard-type integral inequalities, we introduced the notions of $(\alpha, s)$-geometric-arithmetically convex functions and $(\alpha, s, m)$-geometric-arithmetically convex functions, established several integral inequalities of the Hermite–Hadamard type for $(\alpha, s)$-GA-convex and $(\alpha, s, m)$-GA-convex functions, and applied several results in the construction of several inequalities of special means.

**Author Contributions:** Writing—original draft, J.-Y.W., H.-P.Y., W.-L.S. and B.-N.G. All authors contributed equally to the writing of the manuscript and read and approved the final version of the manuscript.

**Funding:** This work was partially supported by the Research Program of Science and Technology at Universities of Inner Mongolia Autonomous Region (Grant No. NJZY20119), China.

**Data Availability Statement:** The study did not report any data.

**Acknowledgments:** The authors thank the anonymous referees for their careful corrections and valuable comments on the original version of this paper.

**Conflicts of Interest:** The authors declare no conflict of interest.

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
