# Peer review of "Hermite–Hadamard’s Integral Inequalities of (α, s)-GA- and (α, s, m)-GA-Convex Functions"

_axioms, doi:10.3390/axioms11110616_

Round 1

Reviewer 1 Report

The authors study the concepts “(α, s)-geometric-arithmetically convex function” and “(α, s, m)-geometric-arithmetically convex function”, and establish some new integral inequalities of the Hermite–Hadamard type for (α, s)-geometric-arithmetically convex and (α, s, m)-geometric-arithmetically convex functions. The results are not considerable and there is no applicable idea in the work. We have to reject it.

Author Response

Read uploaded file

Reviewer 2 Report

Dear Editor

I have read the paper. This paper is regarding the Hermite-Hadamard type of inequalities. Authors have introduced new classes of convexity and derived some related Hermite-Hadamard-like inequalities. There are some suggestions and comments that need to be incorporated before the acceptance of the article.

1) In the introduction section authors have just defined already known classes of convexity. I would suggest writing something about these classes, for example, authors are encouraged to discuss the geometry of geometrically arithmetically convex functions and then discuss the geometry of these newly introduced classes.

2) I also recommend authors say something about their results, and why they are proving these results. What is the significance of these results? In this, my suggestion is to first say something about Hermite-Hadamard's inequality. Its geometrical significance (particularly) and then write about other results. 

3) It would be good to add some applications of the results. 

4) Add conclusion 

Based on these comments my decision is to revise the paper. 

Author Response

Read uploaded file

Reviewer 3 Report

Please check attached file.

Author Response

Read uploaded file
